# Compressive Strength of Modified FRP Hybrid Bars

**DOI:** 10.3390/ma13081898

**Published:** 2020-04-17

**Authors:** Marek Urbański

**Affiliations:** Faculty of Civil Engineering, Warsaw University of Technology, Al. Armii Ludowej 16, 00-637 Warsaw, Poland; m.urbanski@il.pw.edu.pl

**Keywords:** hybrid (HFRP) bars, fiber reinforced polymers (FRP) bars, composite reinforcement, mechanical properties, compressive strength, crushing, buckling

## Abstract

A new type of HFRP hybrid bars (hybrid fiber reinforced polymer) was introduced to increase the rigidity of FRP reinforcement, which was a basic drawback of the FRP bars used so far. Compared to the BFRP (basalt fiber reinforced polymer) bars, modification has been introduced in HFRP bars consisting of swapping basalt fibers with carbon fibers. One of the most important mechanical properties of FRP bars is compressive strength, which determines the scope of reinforcement in compressed reinforced concrete elements (e.g., column). The compression properties of FRP bars are currently ignored in the standards (ACI, CSA). The article presents compression properties for HFRP bars based on the developed compression test method. Thirty HFRP bars were tested for comparison with previously tested BFRP bars. All bars had a nominal diameter of 8 mm and their nonanchored (free) length varied from 50 to 220 mm. Test results showed that the ultimate compressive strength of nonbuckled HFRP bars as a result of axial compression is about 46% of the ultimate strength. In addition, the modulus of elasticity under compression does not change significantly compared to the modulus of elasticity under tension. A linear correlation of buckling load strength was proposed depending on the free length of HFRP bars.

## 1. Introduction

The last 40 years are characterized by an extremely dynamic development of construction materials and a rapid increase in the performance parameters of composite materials. It is said that the engineer has more than 50,000 materials at his disposal. The engineer can and should combine the best properties of these materials to create composites (the most known and widespread is fiberglass) that offer particularly attractive sets of properties [1]. The composite market is large and growing dynamically. Over the next decade, composites will increasingly compete with steel and aluminum in many traditional applications for these metals. FRP materials have excellent mechanical and physical properties, but attention should be paid at the outset to the issue that is usually paramount: Price and availability.

**The economic aspect:** The global composite market is expected to grow from USD 90.6 billion in 2019 to USD 131.6 billion in 2024 [2]. However, the global market for glass and other synthetic fibers will increase from USD 65.9 billion in 2019 to USD 91.4 billion in 2024 (6.8% CAGR) [3]. Fiber reinforced composites currently dominate the composite materials market. In Europe, the production of FRP components (reinforcing bars) using continuous processes such as pultrusion increased by 3.4% in 2018. The total production volume was about 151,000 t (2018), of which carbon fiber accounts for 78,500 t [4]. The price of FRP fibers systematically decreases as demand increases. Carbon fibers are currently manufactured from polyacrylonitrile precursors (PAN). Due to the high cost of these petroleum precursors and associated processing costs, production is expensive. Currently, a low-cost process of producing carbon fiber from lignin is being implemented [5]. The estimated cost of fiber production carbon based on lignin is 6.27 USD/kg and taking into account the shorter thermoset process, the potential cost is reduced to about 4 USD/kg [6].

At present, depending on the type, the price of basalt fibers varies in the range of 2–3 USD/kg. On the other hand, the price of carbon fibers is higher and currently amounts to 10–16 USD/kg [7].

## 2. Novelty and Purpose of the Work

The purpose of the new approach to assess the compressive strength of composite bars is to use them as reinforcement in compressed RC elements (columns), as well as in the compressed zone of bent elements (e.g., beams and slabs). RC compressed elements are the main load-bearing vertical structures of buildings and structures with various functions. They take over the load from floors and shells, transferring them to the foundation and further to the ground under the foundation. Hence, the reinforcement in these elements must have high physical and mechanical indexes and be resistant to corrosion. On the one hand, HFRP bars have a much better combination of mechanical properties than FRP bars with homogeneous fibers. On the other hand, they are corrosion resistant and have significantly better strength characteristics than conventional steel reinforcements.

Empirical research is only possible if it can be reasonably assumed that the examined process is structurally stable. Otherwise there would be no guarantee that small external influences do not significantly change the indications of the measuring instruments. The applied author’s method of anchoring the tested specimens means that the measurement disturbance due to improper mounting is minimized and does not affect the structural stability of the specimens.

The tests carried out according to the proprietary compression test method determine the scope of the application of HFRP reinforcement and its standardization in compressed RC elements. In addition, they allow a rational spacing of transverse reinforcement in compressed RC elements to effectively suppress lateral deformations. At the same time, they allow the optimization of HFRP compressive strength based on the determined correlation strength—free length of a bar.

## 3. HFRP Bars Hybridization Concept

**Characteristics of basalt and carbon fibers**: The properties of basalt and carbon fibers included in HFRP bars have a number of significant advantages, which are presented below.

Basalt fibers have an excellent resistance to acids and alkalis. The weight loss in the environment is three times less than that of glass fibers in the acidic environment [8]. The reduction of tensile strength of basalt fibers at a temperature of 600 °C is only 10% while for glass fibers up to 40% [9]. Bars made of basalt fibers are characterized by an excellent resistance to high temperatures. In turn, tests of tensile strength of basalt fiber bars after 90 min at 300 °C confirmed a much smaller reduction (10%) compared to the GFRP bars (75%) [10,11,12]. However, similar to the GFRP bars, they have a relatively small modulus of elasticity [13].

In contrast, carbon fibers have a high modulus of elasticity and high tensile strength, which are maintained at temperatures exceeding 2000 °C, creep resistance, and relatively high values of tensile strength [14]. Carbon fibers consist of a blend of amorphous carbon and graphite carbon. The high tensile modulus is due to the graphite form in which carbon atoms are arranged in the corners of the connected regular hexagons. There are strong covalent bonds between the carbon atoms in each plane, while the bond between the planes is much weaker than the van der Waals forces, which affects the highly anisotropic physical and mechanical properties of carbon fibers [15].

In addition, carbon fibers have a negative coefficient of thermal expansion (−0.6 to −1.45) ∙ 10^−6^/°C and a relatively high thermal conductivity. These features, combined with high strengths, cause a relatively low susceptibility to thermal shock. Their main disadvantage is the tendency to oxidize at high temperature. Compressive strength decreases with increasing the tensile modulus [16]. However, if the shear modulus (*G*_12_) increases, then the strength of carbon fibers also increases [17].

Bars Hybridization. An HFRP bar is a hybrid that is obtained by using two or more different types of fibers in one matrix. Hybrid bars have a better combination of properties than composites containing only one type of fiber. Various combinations of fibers and matrix materials are used, but it has been decided to combine the beneficial properties of carbon and basalt fibers with a polymer resin. When the HFRP bars are extended, failure is usually not catastrophic (i.e., not sudden). Carbon fibers are the first to fail, and at this point the load is transferred to the basalt fibers. Carbon fibers are strong and relatively stiff and provide a low-density reinforcement; however, they are expensive. Basalt fibers are relatively cheap but do not have carbon stiffness. The basalt-carbon hybrid is stronger and harder, has a higher impact resistance, and can be produced at lower costs than comparable composites reinforced entirely with carbon or entirely with basalt.

The volume fraction of fibers in FRP composites can reach up to 80% [18]. The properties of composite materials, including elasticity modules, were calculated on the basis of the mixture rule (ROM) [19,20]. In HFRP composites, loads are transferred by basalt and carbon fibers, and their strength and modulus of elasticity increases as the volume fraction of fibers increases, according to Equation (1).
*E*_f_ = *V*_m_ ∙ *E*_m_ + *V*_b_ ∙ *E*_b_ + *V*_c_ ∙ *E*_c_ = [1 − (1 + *α*_f_) ∙ *V*_b_)] ∙ *E*_m_ + *V*_b_ ∙ *E*_b_ + *α*_f_ ∙ *V*_b_ ∙ *E*_c_,(1)
where *V*_b_, *V*_c_, *V*_m_—volume fraction of basalt, carbon roving fibers, and polymer matrix, respectively; *f*_b_, *f*_c_, *f*_m_—ultimate strength of basalt, carbon roving fibers, and polymer matrix respectively; *E*_b_, *E*_c_, *E*_m_—modulus of elasticity of basalt, carbon roving fibers, and polymer matrix, respectively; *α*_f_ = *V*_c_/*V*_b_.

However, the modulus of elasticity at the load perpendicular to the direction of the fibers is much smaller and can be described by Equation (2):*E*_f_ = *E*_b_ ∙ *E*_c_ ∙ *E*_m_/{*V*_b_ ∙ *E*_c_ ∙ *E*_m_ + *α*_f_ ∙ *V*_b_ ∙ *E*_c_ + [1 − (1 + *α*_f_) ∙ *V*_b_)] ∙ *E*_b_ ∙ *E*_c_},(2)

In order to increase the bar stiffness, some basalt fibers have been replaced with carbon fibers. The values of the elasticity modulus of HFRP bars were estimated as a property of the additive composite, taking into account the volume shares of basalt and carbon fibers, as well as the properties of the epoxy resin matrix. In this way, the modulus of elasticity was determined for the following ratios of volume of carbon fibers to basalt fibers *α*_f_ = 1:9, 1:4, 1:3, 1:2, and 1:1 (Table 1).

The above analytical considerations based on the Voight model mainly concern the volume fraction of fibers and do not take into account any effect of fiber distribution in the bar cross-section. Therefore, it was decided to perform computer simulations of the fiber arrangement effect using the FEM model for two locations of carbon fibers: Carbon fibers located in the layer adjacent to the bar surface or in the area of the bar core [21]. Figure 1a shows the distribution of basalt rovings for a bar with a nominal diameter of 8 mm. Configurations of basalt and carbon rovings are shown in Figure 1b,c. For the same amount of carbon and basalt fibers, slightly better results were obtained when the carbon fibers were distributed in the surface layer (Figure 1b). It should be noted that the results obtained on the basis of the Voight model for various bar configurations were similar and the differences were about 2% [22]. Due to the local surface burning of carbon fibers during the thermal process of HFRP bars pultrusion, it was decided to place carbon fibers in the core of the cross-section (Figure 1c).

HFRP bars are composed of epoxy matrix and basalt fibers, carbon fibers with a volume ratio of 1:4 (i.e., 16% carbon fibers, 64% basalt fibers, and 20% matrix in the form of a four component 1300 System^®^ epoxy resin, Ciech Sarzyna S.A., Nowa Sarzyna, Poland). Thanks to this substitution, HFRP bars are characterized by a much higher stiffness, which enables more efficient application as reinforcement of concrete elements subjected to bending and compression. The tensile strength of HFRP bars is more than two times higher than the yield stress of steel [23]. However, the compressive strength of HFRP bars has not yet been presented.

## 4. Research Compressive Strength of FRP Bars

HFRP compressed reinforcement is ignored in the standards for the design of RC elements with FRP reinforcement (ACI 440.1R-06) [24]. Similar recommendations can be found in the current ACI 440.1R-15 standard [25]. In turn, the previous Bulletin fib 40 [26] does not recommend the use of FRP bars in the compression zone. CSA S806-12 recognized that for design purposes FRP reinforced elements in the concrete compression zone have zero compressive strength and rigidity. Which is tantamount to not using FRP compression reinforcement [27].

ASTM D695-15 recommends carrying out the test by placing the sample between the cylindrical heads in a compression device. The strength test specimen should have a diameter equal to the diameter of the bar and a length twice that of the bar. The guidelines state that the ends of the sample are cut perpendicular to the loaded axis and as flat as possible [28].

However, this procedure did not take into account crushing and unintentional rotation of the bar at both its ends, which significantly affects the assessment of strength and deformation of the bar under compression. The assessment of compressive strength is necessary to understand and predict the behavior of eccentrically bent and compressed reinforced FRP elements. Despite standard restrictions, recent studies have shown that neglecting the use of GFRP compression reinforcement in concrete columns is too conservative.

Tavassoli et al. examined nine columns reinforced with GFRP bars under cyclic loading and showed that the columns had significant strength and ductility. The failure was initiated by crushing the compressed concrete followed by crushing of the longitudinal reinforcement GFRP. Although this revealed the general applicability of GFRP bars as compression reinforcement in columns, the mechanism of destruction of the bars themselves was still unknown [29].

Afifi et al. showed that the strength of GFRP reinforced columns with sufficient transverse reinforcement spacing is only 7% lower than that similar in structural columns with steel reinforcement [30]. Tests of columns reinforced with GFRP bars were carried out by Tobi et al. They showed that the GFRP bars used had a 10% share in the column load capacity, which is close enough to the steel bar share (12%). This proves that the GFRP bars can be used in compressed elements, provided that there is an appropriate restriction to eliminate bar buckling [31].

Based on the research, it can be concluded that compressed elements with composite reinforcement should be designed by other methods than compressed elements with steel reinforcement. Different spacing of transverse reinforcement should be used in comparison with steel reinforcement. Currently, it is recommended to ignore the share of FRP fibers along the axial compression force of concrete elements due to the lack of conclusive data on this topic [32].

Tests on FRP bars with a length to diameter ratio of 1:1 to 2:1 showed that the compressive strength is much lower than the tensile strength [33]. Failure methods of FRP bars due to compression in the free area of the (nonanchored) bar are identified as crushing, shearing, and buckling. In addition, the method of bar destruction also depends on the type of fiber, the volume fraction of the fiber in the bar, and the type of resin. The compressive strength is higher for bars with a higher tensile strength. The modulus of elasticity under compression is smaller than the modulus of elasticity under tension (about 80%) [15,34].

Fujisaki and Kobayashi tested FRP bars embedded in concrete anchors at both ends of the sample with a free length of 5 mm under compression. The compressive strength of the AFRP, CFRP, and GFRP bars was determined as 10%, 30–50%, and 30–40% tensile strength. However, the free unbraced length between two concrete anchorages of samples with diameters from 6 to 10 mm was only 5 mm [35]. This caused the sample to be in a complex state of stress, which in turn led to a significant reduction in the compressive strength.

Dietz et al. conducted tests of 45 GFRP reinforcing bars, 15 mm in diameter and 50–380 mm free (unbraced) length to determine their ultimate strength and compressive modulus of elasticity. Three different failure modes were observed, namely crushing, buckling and crushing, and buckling. Failure modes were dependent on the free length of the samples. Individual samples were mounted in two 135 mm long steel cylinders with an external diameter of 50 mm, intended for screwing into the UTM heads of the testing machine. A hole 65 mm long and 17.5 mm in diameter was drilled in the center of each threaded cylinder so that any possible rotation of the sample ends would not lead to uncertainty of the results. GFRP samples with a ratio of free bar length to diameter *L*_f_/*d*_b_ < 7.33 (*L*_f_ < 110 mm) were destroyed by crushing. The second failure mode for samples with 7.33 < *L*_f_/*d*_b_ < 14 of free length was a combination of crushing and buckling failures. For samples with free length of *L*_f_ > 210 mm (*L*_f_/*d*_b_ > 14), the failure mode consisted of almost clean buckling of the free length bar. Samples showed a wide spread due to crushing, while those which were damaged due to buckling showed a small spread of values. Based on the limited tests of the three samples, it was found that the modulus of elasticity under compression is approximately equal to the modulus of elasticity under tension. In addition, it was found that for unbuckled reinforcing bars with a free length of *L*_f_ < 110 mm, the ultimate compressive strength is about 50% of the final tensile strength. [36]. Due to the improper anchoring of the samples, the results for free lengths *L*_f_ < 10*d*_b_ are characterized by too large a dispersion, which makes it difficult to infer as to the value of the results.

Brunn tested 34 samples of 25 mm GFRP bars for compression. Specimens of GFRP bars were cut perpendicular to the longitudinal axis and then placed in UTM holders. The specimens were loaded at a rate of about 100 MPa/min. The tested samples had a free length *L*_f_ from 2 to 40 bar diameters (50 to 600 mm, respectively). It was found that GFRP bars with a *L*_f_/*d*_b_ < 9.2 (*L*_f_ < 230 mm) are damaged by crushing, while with a *L*_f_/*d*_b_ > 12.6 (*L*_f_ > 315 mm) the failure is carried out due to buckling. The mechanism of failure in the transition zone of 230 mm < *L*_f_ < 315 mm was not recognized [37]. Since the ends of the specimens may have been crushed locally, the results may have been prone to varying results. It should also be determined whether a similar behavior can be observed in samples of bars with smaller diameters.

Khan et al. performed the tensile test of three GFRP and three CFRP bars in addition to five CFRP bars and five GFRP bars with 15.9 and 15.0 mm in diameter and 80 and 60 mm, respectively. Each compression specimen with flat parallel ends was placed directly and vertically between the UTM heads and tested under the load with controlled displacement at a speed of 1.0 to 1.3 mm per min until damaged. The compressive strength values obtained for the GFRP bars were 1.4 times higher than the CFRP bars, and the final compressive stress corresponding to the final compressive strength was 1.65 times higher in GFRP bars than in CFRP bars. The final compressive stress of the tested FRP bars obtained from the tests were reported as registered in the compression test because there is no slip and the values are reliable. The compression modulus of elasticity of CFRP bars was 1.17 times greater than that of GFRP bars. The modulus of elasticity at compression of GFRP bars was 42.0 GPa and was close to the value of the module given by Dietz [38].

Khorramian and Sadeghian studied GFRP bars with a diameter of 13 to 21 mm in a modified design device. The compressed specimen was mounted in a device consisting of two steel plates, two steel rings, two anchor adhesives, and a GFRP bar. The steel rings had a depth equal to the diameter of the bar, a diameter twice as large as the diameter of the GFRP bar, and a minimum thickness of 3.3 mm. The GFRP bar was then fastened with a fast-drying adhesive. It has been proposed that the total length of the GFRP bar is four times larger than its nominal diameter, which means that the free length of the bar is 2 diameters [39]. This length is too short to avoid the effect of fastening one end of the bar to the other. This situation affects the correct estimation of compressive strength. In addition, despite diligence in aligning the sample tips, local bar crushing may occur.

## 5. Failure Mechanism and Microscopic Analysis of Compressed HFRP Bars

Failure Mechanism. To properly assess the effect of compressive forces on the behavior of HFRP bars, you must first know the mechanism of destruction of FRP composites. The compressive strength of composites is less than the tensile strength. This is due to the fact that the fibers undergo buckling, or more precisely—they break down—a type of simultaneous buckling, shown in Figure 2b. Thus, while brittle ceramics are resistant to compression, the composites exhibit a much better tensile strength.

During compression of composites with a parallel (quasi-parallel), relative to the load, orientation of the fibers often occur in their elastic micro-buckling (Figure 2b), which under the influence of stresses cause formation of critical micro-buckling (plastic micro-buckling) related to fiber breaking. Analyses of flexible and plastic kinking of micro-buckling are able to take into account some, but not all, experimental observations. Fiber bending must be treated unequivocally, so that you can predict the width of the micro-buckling band and its angle of inclination [40].

The mechanism of this type of failure in the form of micro-buckling in FRP composites is most often associated with a change in the direction of crack propagation at the fiber-matrix phase, in which the gap changes direction, spreading deeper into the matrix. At the initial stage of loading, elastic bending of FRP fibers occurs (Figure 3a). Then, as the stress increases, the deformation of the fibers goes to the state of elastic micro-buckling (Figure 3b). After exceeding the stress limit, FRP fibers in the matrix reach the state of plastic micro-buckling which leads to local fiber kinking (Figure 3c) [41].

**Microscopic Analysis:** Before starting the compression tests, it was decided to conduct a microscopic analysis to determine the effect of chemical composition on the properties of HFRP bars.

The elemental composition analysis using a secondary X-ray energy dispersion spectrometer (EDS) provided important information on the composition of HFRP bars. While carbon fibers are homogeneous in terms of chemical composition, the chemical composition of basalt fibers has a significant impact on the strength properties of HFRP bars. SiO_2_ silica plays the dominant role in the chemical composition of basalt fibers, constituting approximately 43% by weight, the Al_2_O_3_ content is 17%, while FeO, CaO, and MgO are present in comparable amounts of several percent. Unlike glass fibers, basalt fibers contain Fe_2_O_3_ and FeO, which affect density (2.73 g/cm^3^), strength, and are responsible for the olive-brown color of the fibers (Figure 4).

By examining HFRP bars using the BSE (backscattered electrons) method, significant data on the configuration of basalt and carbon fibers inside the samples was obtained. Figure 5a presents a SEM BSE image with visible broken basalt fibers (B) and carbon fibers (C)—visible difference in diameter of basalt and carbon fibers. On the other hand, in Figure 5b, visible micro-buckling with fracture of carbon fibers is shown.

Breakage of the main load carrier of the composite structure (fibers) is considered the final state in the process of its damage (Figure 5b). It leads to the formation of a so-called failure path through local, previously damaged zones, in consequence causing loss of the structure’s ability to carry loads through its physical destruction [42].

## 6. HFRP Bar Compression Test Method

Due to the free length of longitudinal FRP bars that may vary in the column depending on the spacing of the spiral or stirrup chosen by the designer, it is necessary to establish the relationship between the buckling load strength and the nonanchored length for these bars. Therefore, it is extremely important to optimize transverse reinforcement spacing, which significantly affects the possibility of local buckling of FRP longitudinal reinforcement. Therefore, it was decided to carry out buckling load strength tests for HFRP bars based on the newly developed compression test method. HFRP bars with six different free lengths of five pieces were tested. The term “free length—(*L*_f_)” means the bar length between the sample anchorages. The variation in free bar lengths reflects the behavior of composite longitudinal reinforcement limited by the stirrup spacing. By specifying the buckling load strength for various free bar lengths, it is possible to optimize the transverse reinforcement spacing depending on the type of FRP longitudinal reinforcement.

**Sample preparation:** The total length of the samples of each of the samples made of HFRP bars with an equivalent diameter of *d*_b_ = 8.77 mm was *L* = *L*_a_ + *L*_f_ + *L*_a_. The tested HFRP bars were manufactured from the same lot to ensure uniform material properties (Figure 6a). The size *L*_a_ is the length of the anchoring of the bar at both its ends and in value it was equal to the longitudinal dimension of 120 mm long steel sleeves. Since the unanchored length of the longitudinal reinforcement bars may differ in the columns depending on the spacing of the spiral or stirrups, it is necessary to determine the relationship between the buckling load strength and the unbraced length of the bars. The value of the measuring length (free sample lengths—nonanchored) was times the diameter of the tested bars and was *L*_f_ = *n* ∙ *d*_b_. The tests were carried out for another six values of *n* = 6, 10, 14, 20, 22, 26. The steel sleeves had a length of 120 mm and an external diameter of 42 mm. The test bars were centrally attached to the sleeves after degreasing with a special adhesive with a hardener. First, the bar was anchored in one sleeve and then, after the connection was cured, the other end of the sample was anchored (Figure 6b). In each sleeve 16 mm diameter holes have been made. The holes in the sleeves were slightly larger than the diameter of the bars tested. In order to avoid an unplanned crushing inside the sleeve, holes have been made along their entire length. Compression forces are transferred due to the adhesion between the side surface of the anchored rod and the inner surface of the sleeve. Adhesion is ensured by a specially developed resin mix with the addition of fixing components. The effectiveness of the anchorages was each time monitored immediately after the sleeve ends. Then, in the half free length of each sample, two strain gauges were mounted on opposite sides of the tested bar. Before installing the strain gauges, the surface of the bar was ground and cleaned so that the deformation measurements were burdened with the smallest possible error.

The designed anchorages in the sleeves are designed to absorb the compressive force of the test bar through its lateral surface. Thanks to this solution, there is no degradation of the bar ends, and the rod itself does not move during the test, which may result in premature buckling. In addition, there is no need to check the perpendicularity of the bar end surface to its longitudinal axis.

**Measuring procedure:** The test sample must be correctly placed in the center of the test machine so that it is aligned with the machine in order to obtain a clean compression load. The configuration diagram of the sample anchored on both sides is shown in Figure 7a. To properly position the sample, a centering tube with a length of 450 mm and an internal diameter of 44 mm was used to allow free movement of the sample anchored on both sides (Figure 7b). Openings were cut out in two opposite sides along the longitudinal axis to allow measurement and observation of the sample.

In addition, to avoid accidental eccentrics, the tube with the anchored sample was placed at the bottom in a round centering plate, which allowed a low rotation and self-centering operation (Figure 8a). It should be emphasized that the weight of the sleeve was omitted because the corresponding load was very low compared to the compressive load applied. Figure 7b shows one of the samples placed in the testing machine. Data was recorded using two RL20 (Techno-Mechanik, Gdańsk, Poland) strain gauges with the length of 30 mm, a gauge length of 20 mm, and a gauge resistance of 120 Ω. The extensometer size was chosen based on availability for the study group. The strain gauges were installed longitudinally halfway along the opposite sides of the rod being tested. The arithmetic mean of the deformation readings was used to calculate the modulus of elasticity.

Tests of the transverse buckling load strength of HFRP bars were carried out in accordance with the following assumptions. In order to obtain consistent and repeatable results, the speed of the load increase was important. Due to the high impact of buckling load strength since loading. In the case of compression tests, the load degree was generally defined at about 100 MPa/min, so that the destruction of samples occurred before 7 min. If the destruction of the bar occurred outside the free length zone, additional testing of a sample was taken from the same batch. The load was implemented continuously and gradually to avoid impact. The tested HFRP bars had a spiral braid made of basalt fiber. The bars were tested in a ZD20 (WPM, Leipzig, Germany) testing machine with a maximum load of 200 kN. Sampling was performed at 10 measurements per second including axial strain, axial load, and axial stroke displacement from the machine. The study was considered complete just after obtaining the maximum compressive force.

## 7. Results and Analysis

The compressive strength of HFRP bar samples is determined according to the Equation:*f*_f,c_ = *F*_u_/*A*_f,min_,(3)
where *F*_u_ is the maximum force registered during the test; *A*_f,min_ is the minimum bar diameter measured immediately before installing the strain gauges on the surface of the bar being tested.

The influence of shear load and radial tensile load occurring in the intermediate zone from the test sleeve to the bar on the sample destruction process is not taken into account. During the study, three different failure modes of HFRP bar samples were observed:Bar failure occurred due to the shear of the bar for smaller free lengths. This destruction occurred as a result of splitting individual fibers and separating them from the epoxy matrix 6*d*_b_; (Figure 9a).In the samples in which the free length was 22 and 26*d*_b_, the failure occurred as a result of buckling combined with the delamination of individual fibers of the tested bar. Ultimately, the failure occurred as a result of cracking of some fibers and detachment of the epoxy matrix (Figure 9c).In samples of intermediate length (10*d*_b_, 14*d*_b_), the onset of failure occurred as a result of buckling, and with further loading, the fibers were simultaneously split and the tested bar crushed (Figure 9b).

Figure 9a shows an HFRP bar with a free length of 6*d*_b_, which was damaged due to crushing on a free length section. The failure of samples was usually in the middle of the free length of the bar. The stress strain diagrams in both cases were analogous and no significant differences were observed in the values of the modulus of elasticity and buckling load strength. HFRP samples with a free length of 14*d*_b_ were destroyed as a result of initially buckling and then crushing (Figure 9b). The third failure mode of HFRP bars with a free length greater than 14*d*_b_ that consisted of buckling the sample at a maximum compressive force but without crushing, is shown in Figure 9c. It should be noted that for some samples with a free length above 20*d*_b_ at a maximum load, there were no signs of visible damage on the bar surface. Based on the test results, the dependence of buckling load strength on the free length of bars was developed.

**Stress****–strain relationship:** The stress–strain relationship for compressed bars was constructed by determining the stresses in accordance with Equation (3) and determining the strain as the mean strain value recorded from two strain gauges attached on opposite sides in the middle of the sample. The average buckling load strength for samples with successive values of free lengths and their standard deviation and the coefficient of variation are presented in Table 2. The buckling load strength of HFRP bars decreases with the increase of the free (nonanchored) length. The buckling load strength was compared with the tensile strength of HFRP bars with the same nominal diameter of 8 mm. The buckling load strength is 46% of the tensile strength (for 6*d*_b_ free length) and decreases to 15% as the anchor length increases (for 26*d*_b_ free length) (Table 2, column 4). In Table 2, column 5, the average deformation values for samples with successive nonanchored lengths are shown. During compression of the deformation (*ε*_uc_), in terms of absolute value, they reach from 1/3 to 2/3 of the tensile deformation (*ε*_ut_), with no apparent effect of free length on the deformation value (Table 2, column 7).

For HFRP samples with a free length of 6 and 10*d*_b_, the stress–strain curves continued until crushing, and the load suddenly decreased after the failure. The bar failure occurred only on the free length of the samples. In addition, the stress–strain relationships were linear and were characterized by high consistency of results (Figure 10a,b). As the 14*d*_b_ free length increased, the indications of opposite strain gauges were increasingly divergent (Figure 10c). Samples with an unbraced length greater than 20*d*_b_ buckled without clear signs of kinking during the failure load (Figure 10d–f). Relations between stress and absolute strain values for subsequent free lengths of HFRP bar samples are shown in Figure 10.

In the case of continuous fiber reinforced composite materials, the elastic modulus can be large and the linear part (where the elastic modulus is constant) of the stress–strain curve is not negligible. For this nonlinear behavior, the tangent or secant modulus of elasticity is usually used. The average values of the modulus of elasticity are shown for specimens of successive free (unbraced) lengths (Table 2, column 8). The modulus of elasticity under compression has a value similar to or slightly higher than the modulus of elasticity under tension (Table 2, column 12). For specimens with free lengths greater than 14*d*_b_ due to buckling, the strain readings differ significantly diametrically, which affects the proper estimation of the modulus of elasticity (Figure 10d–f). Specimens with free lengths greater than 20*d*_b_ showed a greater spread of results compared to samples with short free lengths, which were damaged by crushing.

The modulus of elasticity under compression for subsequent HFRP bar specimens can be determined in several ways. The first way is to determine the modulus of elasticity under compression based on procedure B2 of the ACI 440.3R-12 [43] standard used for tension bars. Since the behavior of the samples as a result of the load and strain shows linearity, the modulus of elasticity *E*_fc_ (Table 2, column 8) was determined by dividing the difference in stress by the difference in strain for the values of 0.5 and 0.2 of the ultimate force. The modulus determined in this way is the tangent of the incisal inclination running through the two mentioned points. Another way is to calculate the average modulus of elasticity in compression *E*_fc,ave_ (Table 2, column 9) based on subsequent readings of stress and strain in the range from 0.2 to 0.5 of the ultimate force. The modulus determined in this way usually reaches values slightly higher according to the first procedure. The third way is to assess the minimum value of the tangent modulus of elasticity *E*_fc,min_ (Table 2, column 10) in the range from 0.2 to 0.5 of the ultimate force. In the initial load range (up to 0.2 ultimate force) the modulus values are unstable due to the sample being accommodated under the load. On the other hand, the modulus values at the load above 0.5 ultimate force are subject to high uncertainty due to the average strain measurement. The modulus of elasticity values *E*_fc_ and *E*_fc,ave_ determined in accordance with the first two procedures, due to the small variability indicate the coherence of both methods for determining the modulus of elasticity. However, the minimum compressive tangent modulus of elasticity is usually achieved not for an initial value of 0.2 ultimate force (as in the case of stretching), but for end values in the range from 0.2 to 0.5 of the ultimate force.

For subsequent unanchored lengths, the minimum value of the compressed tangent modulus of elasticity varies from 0.95 to 1.00 of the maximum value (for 0.2 ultimate force) (Table 2, column 11). The relationship of the tangent modulus of elasticity on stress for short and long free lengths of samples is shown in Figure 11a,c, respectively. The tangent tensile modulus *E*_ft_ for the same HFRP bar diameter increases slightly as the load increases. The comparison of the tangent tensile modulus of elasticity *E*_ft_ and tangent compressive modulus of elasticity *E*_fc_ for the whole range of absolute stress values (from 0.1 to 1.0 ultimate stress) *f*_ut_ and *f*_uc_, respectively is shown in Figure 11b. For all HFRP samples, the tangent modulus of elasticity under compression decreased with the increasing load. This phenomenon is associated with the mechanism of micro-buckling, which was described in the previous chapter. The tensile tangent modulus increased as the load increased. This behavior is understandable due to increased stiffness when tensioning HFRP bars. According to the author, the minimum tangent modulus of elasticity *E*_fc,min_ is the proper (best) measure of the stiffness of the tested bars due to compression, which should be taken into account when designing due to compression. In secondary structural elements, the compressive modulus of elasticity of HFRP bars can be determined based on the modulus of elasticity in tension. However, determination of the buckling load strength based on the compression test may be necessary, due to the large disproportion of the buckling load strength depending on the unbraced free length of HFRP bars.

**Correlation of buckling load strength and free rod length:** Under ideal real conditions, the bars always have some manufacturing inaccuracies, while the load can work obliquely or load the bars eccentrically. Under these conditions, buckling caused by compression exists practically from the very beginning of the deformation. However, its effect is small until the values of forces become critical. As the length of the free anchored bars increases, the loss of stability occurs in the elastic working range of the material. The buckling load strength of HFRP bars was determined for each unbraced free length of the sample, the relationship of which is shown in Figure 12. The first stage consisted in determining the correlation between the buckling load strength values and the free length of HFRP bars. The question arises: How can we determine whether the estimated regression equation reflects the well-mentioned relationship. One measure of regression matching is the determination coefficient *R*^2^. According to the trend line calculated by the method of least squares of errors (SSE), the value of the coefficient of determination is 0.9904. This means that the vast majority of the variability of buckling load strength is highly correlated with the change in the length of the unbraced HFRP bars (Figure 12).

Standard deviations of the buckling load strength for subsequent nonanchored bar lengths are marked. In addition, three zones were determined, differing in the failure mode depending on the length of the nonanchored bars. The destruction of samples in zone 1 occurs by crushing the bars without buckling. In transition zone 2, the samples are destroyed due to the initial buckling and then the crushing of the bars. The destruction of bars in zone 3 occurs only as a result of buckling, often without visible signs of cracking of the bars. The stress–strain relationship, shown in Figure 12, is linear in all three of these zones and is well corresponded to the test results. Based on the graph, it is possible to choose the transverse reinforcement spacing depending on the designed buckling load strength, as well as the selection of the appropriate cover thickness due to the bar failure mode.

## 8. Conclusions and Summary

The research showed that the optimal free length of the HFRP bar can be found for which the buckling load strength reaches its maximum value. This is of practical importance due to the determination of the optimal spacing of stirrups in RC compressed elements.

A favorable effect of substitution of some basalt fibers by carbon fibers in HFRP bars was found. In addition to the increased modulus of elasticity under compression, compressive strength increased by 55%; e.g., for 6*d*_b_ from 375.7 to 583.6 MPa for BFRP and HFRP, respectively.

The stress–strain relationship of HFRP bars is different (straight line) from the three-phase relationship for BFRP bars of a similar diameter.

The buckling load strength monotonically decreases with the increasing length of the nonanchored HFRP bar according to the linear relationship established with a correlation of *R*^2^ = 0.99. This allows a simple and reliable selection of the transverse reinforcement spacing depending on the effort of compressed HFRP bars.

The compressive tangent modulus of elasticity in the range of 0.2–0.5 the ultimate load decreases by 4 GPa (5%) in contrast to the tangent elastic modulus at tension, the value of which increases in the analogous range for the ultimate load at 4 GPa (2.5%).

A representative value of the modulus of elasticity (according to the author) should be the minimum value of the tangent modulus of elasticity for the range of 0.2–0.5 ultimate load. The minimum tangent modulus of elasticity *E*_fc,min_ is the correct (best) measure of the stiffness of the tested bars due to the buckling load, which should be taken into account when designing due to compression.

The values of the modulus of elasticity under compression (*E*_f,c_) for nonbuckled bars are similar to the modulus of elasticity under tension (*E*_f,t_). For bars with larger free lengths, the tangent modulus of elasticity under compression reached slightly higher values.

According to the author, HFRP bars can be used as compressive reinforcement, but under the following specified conditions:

Further research should be carried out to determine the relationship of the buckling load strength–free length for the remaining HFRP bar diameters used to propose a general relationship for the design of reinforcement in the compressed zone.

An important factor affecting the reduction of the longitudinal compressive strength of HFRP bars is the misalignment of basalt and carbon fibers or their bending.

Factors that improve the longitudinal compressive strength of HFRP anisotropic bars are: Increasing the value of the matrix shear modulus, tensile modulus, and thus compression, limit deformation and matrix interfacial strengths.

## Figures and Tables

**Figure 1 materials-13-01898-f001:**
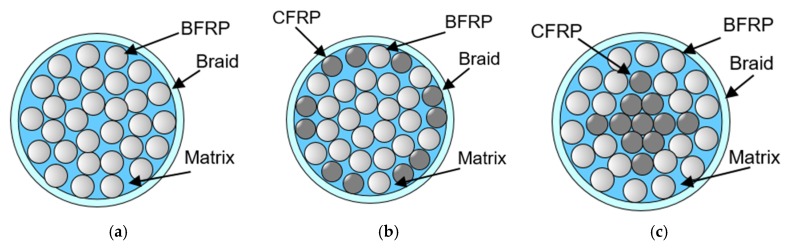
Cross-section types of a fiber reinforced polymer (FRP) bar: (**a**) Basalt FRP (BFRP) bar—only basalt rovings; (**b**) HFRP bar—carbon rovings at the surface of the cross-section; (**c**) HFRP bar—carbon rovings in the core of the cross-section.

**Figure 2 materials-13-01898-f002:**
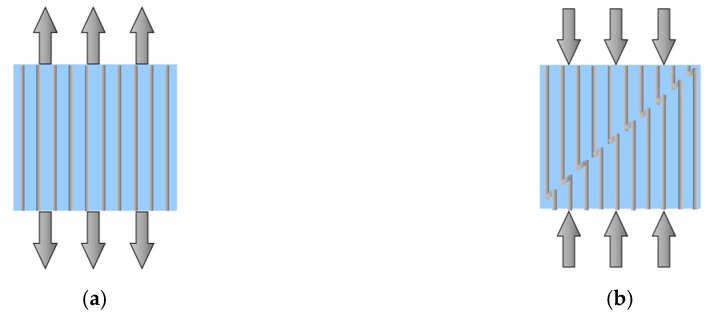
FRP bars: (**a**) Tension mode; (**b**) compression mode, failure by kinking of fibers.

**Figure 3 materials-13-01898-f003:**
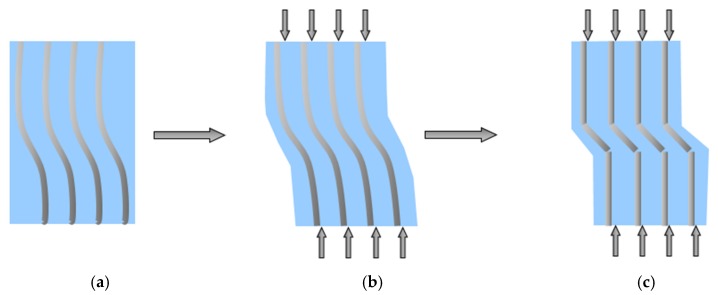
Mechanism of fiber micro-buckling and breaks in the composite: (**a**) Buckling in the 0° plane with initial fiber misalignment; (**b**) elastic micro-buckling of compression-loaded fibers; (**c**) fibers kinking phenomena causing catastrophic fracture.

**Figure 4 materials-13-01898-f004:**
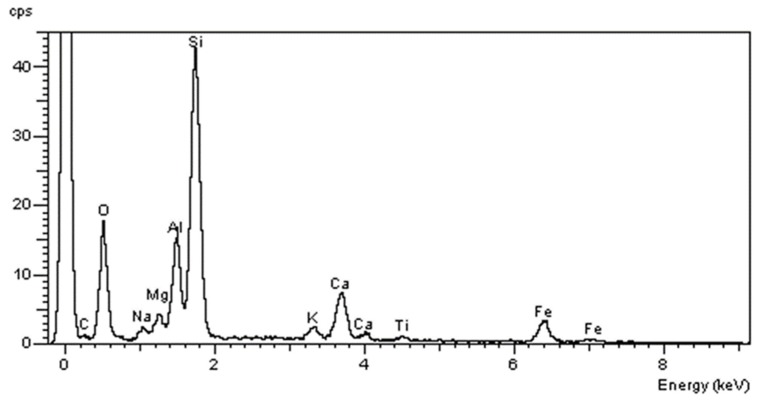
Spectrographic analysis of basal fibers of the HFRP bar (own research).

**Figure 5 materials-13-01898-f005:**
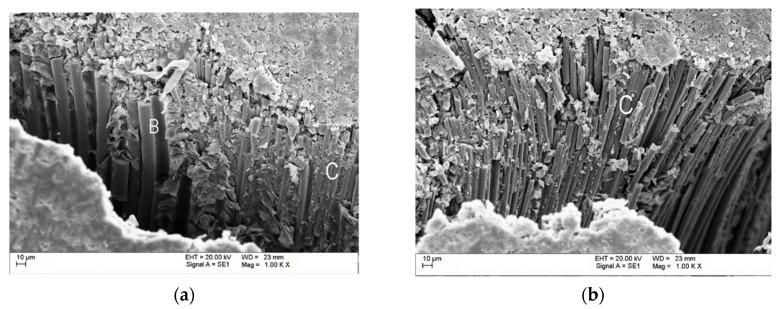
HFRP bar with a diameter of 8 mm: (**a**) Visible broken basalt (B) and carbon (C) fibers; (**b**) micro-buckling with broken carbon fibers (C); (own research).

**Figure 6 materials-13-01898-f006:**
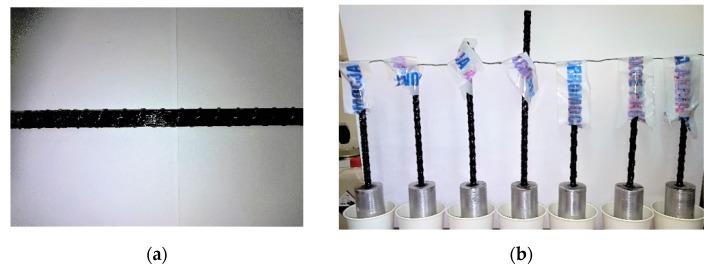
Sample preparation procedure: (**a**) The sample size for testing; (**b**) one-sided anchoring of the bars in a steel sleeves.

**Figure 7 materials-13-01898-f007:**
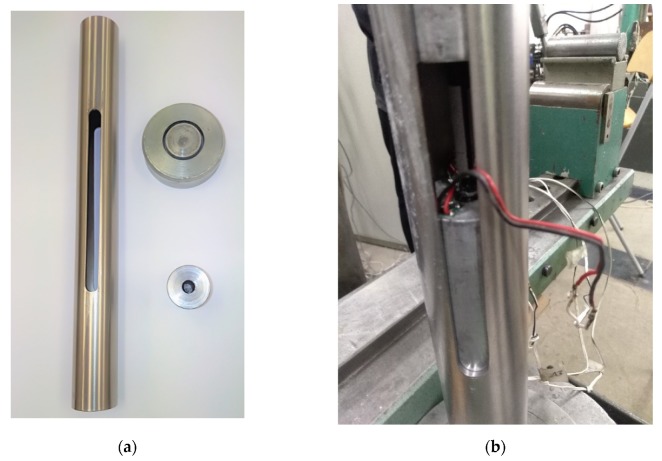
Testing device: (**a**) Components in the bar compression test; (**b**) openings on the lateral surface of the centering tube.

**Figure 8 materials-13-01898-f008:**
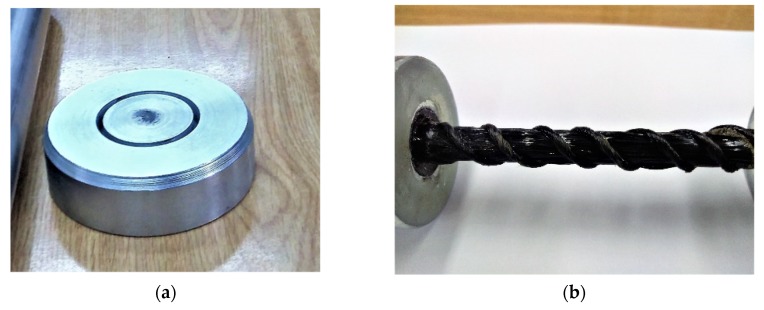
Attachment of the samples: (**a**) Lower circular steel plate for centering the sample; (**b**) anchor sleeves.

**Figure 9 materials-13-01898-f009:**
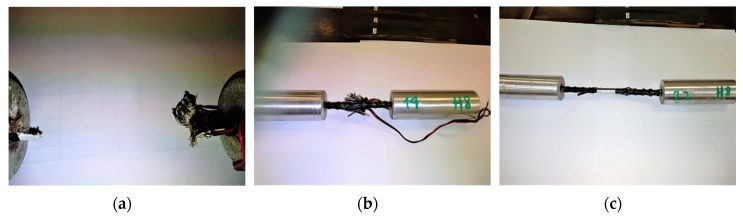
Failure mode of HFRP bars with a diameter of 8 mm: (**a**) Crushing the sample with *L*_f_ = 6*d*_b_; (**b**) buckling and crushing of the sample with *L*_f_ = 14*d*_b_; (**c**) buckling of the sample with *L*_f_ = 22*d*_b_.

**Figure 10 materials-13-01898-f010:**
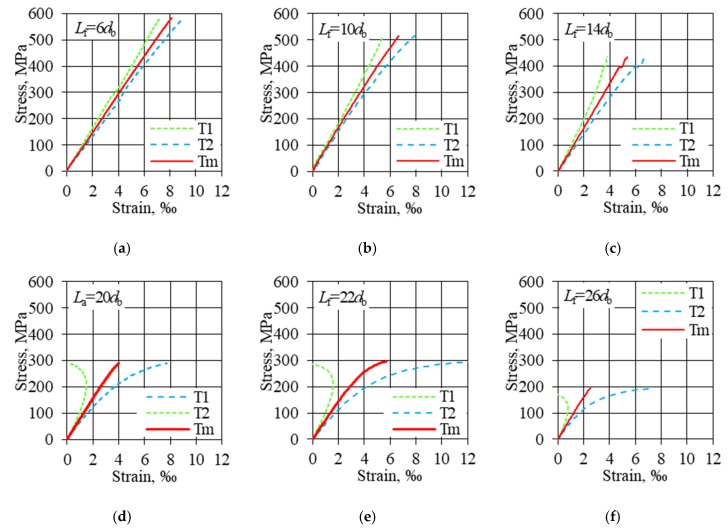
Stress–strain relationship (absolute value): (**a**) *L*_f_ = 6*d*_b_ crushing of the specimen; (**b**) *L*_f_ = 10*d*_b_ crushing the sample; (**c**) *L*_f_ = 14*d*_b_ initial buckling followed by crushing; (**d**) *L*_f_ = 20*d*_b_ buckling without visible crushing; (**e**) *L*_f_ = 22*d*_b_ buckling without crushing; (**f**) *L*_f_ = 26*d*_b_ buckling without visible crushing.

**Figure 11 materials-13-01898-f011:**
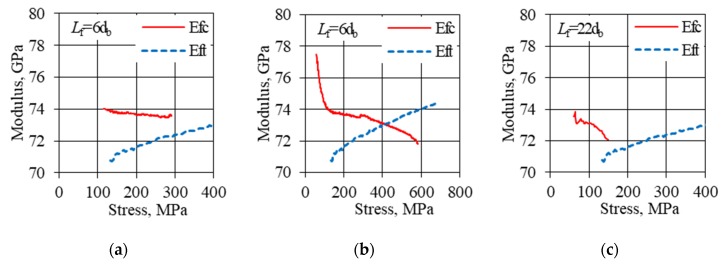
Comparison of the tangent modulus of elasticity at tensile *E*_ft_ in the range of (0.1–1.0)*f*_ut_ with the tangent modulus of elasticity in compression: (**a**) *L*_f_ = 6*d*_b_, *E*_fc_ in the range of (0.2–0.5)*f*_uc_; (**b**) *L*_f_ = 6*d*_b_, *E*_fc_ in the range of (0.1–1.0)*f*_uc_; (**c**) *L*_f_ = 22*d*_b_, *E*_fc_ in the range of (0.2–0.5)*f*_uc_ (absolute stress value).

**Figure 12 materials-13-01898-f012:**
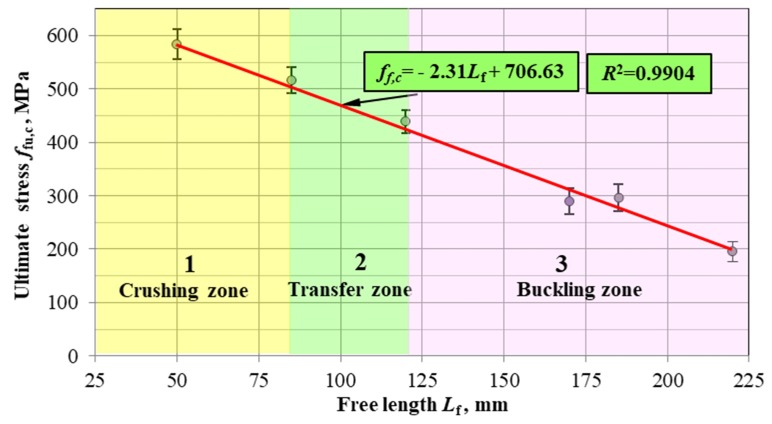
Proposition of the linear relationship of the ultimate buckling load strength on the free length (not anchored) for HFRP bars with a nominal diameter of 8 mm.

**Table 1 materials-13-01898-t001:** Tensile modulus of elasticity of hybrid fiber reinforced polymer (HFRP) bars.

Fiber Volume	Modulus of Elasticity of HFRP Bars, GPa		
Ratio	ROM Analysis	FEM Analysis	
Carbon/Basalt	Carbon Fibers Location		
*α* _f_	neglected	at the surface	in the core
1:9	83.3	83.0	82.9
1:4	94.8	93.6	93.6
1:3	100.5	101.1	100.1
1:2	110.0	110.5	109.1
1:1	129.1	127.8	129.2

**Table 2 materials-13-01898-t002:** Comparison of compression and tension characteristics of HFRP bars (absolute value).

*L* _f_	*f* _fc_	CoV	*f*_fc_/*f*_ft_^1^	*ε* _uc_	CoV	*ε*_uc_/*ε*_ut_^1^	*E* _fc_	*E*_fc_,_ave_	*E* _fc,min_	*E*_fc,min_/*E*_fc,0.2_	*E*_fc_/*E*_ft_^1^
mm	MPa	MPa	–	‰	‰	–	GPa	GPa	GPa	–	–
1	2	3	4	5	6	7	8	9	10	11	12
50	583.60	28.13	0.46	9.01	0.42	0.52	73.37	73.68	73.46	0.99	0.99
85	516.43	23.55	0.40	6.06	0.30	0.35	78.67	82.75	81.08	0.95	1.10
120	439.12	20.99	0.34	6.95	0.31	0.40	83.89	82.41	81.65	1.00	1.10
170	289.48	23.85	0.23	7.73	0.62	0.45	78.18	78.22	77.75	1.00	1.05
185	296.18	25.74	0.23	12.49	1.15	0.72	71.11	73.02	72.07	0.98	0.98
220	195.68	18.86	0.15	7.31	0.72	0.42	81.26	82.34	81.55	0.99	1.10

^1^ The average tensile strength of HFRP bars with the same diameter, the ultimate strain, and the tensile modulus of elasticity were *f*_ft_ = 1277.92 MPa, *ε*_ut_ = 17.3‰, and *E*_ft_ = 73.89 GPa, respectively (own research).

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
