# Peer review of "Compressive Strength of Modified FRP Hybrid Bars"

_materials, 2020, doi:10.3390/ma13081898_

Round 1

Reviewer 1 Report

A lightweight carbon/basalt hybrid composite bar was developed to replace the steel rebar for reinforced concrete and test the compressive strength.

  1. Concrete is strong under compression and the rebar used to increase the tensile strength of the RC (tensile face under the bending moment). Why the compressive strength of the rebar is important? If you want using the composite rebar to improve the compressive strength of the concrete, compressive strength of the hybrid composite bar should be higher than the concrete. 
  2. What is the CTE of the hybrid composite bar? Thermal residual stress may be caused by the difference between concrete and composite bar CTE. Is this not a problem for RC safety? Moreover, because the carbon fiber and basalt fiber have different thermal expansion coefficient, thermal residual stress already remain in the hybrid bar (temp. differnce between epoxy curing temp. and working temp.) 
  3. Eq 1. suggest the ROM for tensile strength. However, this prediction is very inaccurate method with low. There are sevearal modified methods (ex, modification of rule of mixtures for tensile strength extimation of circular GFRP rebars, polymers, 9(12): 682, 2017) Plz check it.
  4. Strength and stiffness should be constant regardless of the length. As the author mentioned, the load measured by the compression test (data of buckling zone in Figure 12) is a buckling load, not compressive strength. And the buckling load following the equation P = π2*E*I/L2. Only a short specimen (data of the crushing zone) is considered to be the compressive strength. Plz check the standard method to measure the compressive strength (ex, astm d695-10, d6641 or d3410)
  5. Does the cylindrical jig of Figure 7-(a), not interfere with the buckling of the specimens?
  6. The actual rebar will not have a free length surrounded by concrete, what does it mean to take free length as a main test variable?
  7. It is judged that the slope of the strain varies depending on the applied stress as a result of the material being damaged or deformed. Can it be represent that the stiffness has changed as described in Figure 11.?

Author Response

At the beginning I would like to thank you for the insightful and accurate comments contained in the commentary to the article. Below I will try to answer the issues raised and also adapt to the comments in the article.

Point 1: Concrete is strong under compression and the rebar used to increase the tensile strength of the RC (tensile face under the bending moment). Why the compressive strength of the rebar is important? If you want using the composite rebar to improve the compressive strength of the concrete, compressive strength of the hybrid composite bar should be higher than the concrete.

Response 1: A compressed zone exists in compressed and bent RC elements. Reinforcement in the compression zone contributes to the increase of compressive strength of the element without the need to increase its cross-section. It is obvious that the compressive strength of the composite bar should be greater than the compressive strength of the concrete. Otherwise, these bars would weaken the cross-section. It is important to establish a safe spacing of the stirrups to exclude local buckling, which can cause spalling of the concrete cover. This phenomenon also occurs in bent elements, e.g. in beams. I observed this phenomenon while testing RC beams with composite reinforcement. In this case, the correct spacing of the stirrups is also important, as it will prevent buckling of the bars. That is why it is so important to determine the relationship of the buckling load on the non-anchored bar length, which was the purpose of the research.

Point 2: What is the CTE of the hybrid composite bar? Thermal residual stress may be caused by the difference between concrete and composite bar CTE. Is this not a problem for RC safety? Moreover, because the carbon fiber and basalt fiber have different thermal expansion coefficient, thermal residual stress already remain in the hybrid bar (temp. differnce between epoxy curing temp. and working temp.)

Response 2: The CTE value for the HFRP bar has not been determined during the tests. The HFRP bar consists of 64% basalt fibers (total bar volume), which have a similar CTE ratio to concrete (basalt → 9 ÷ 12 x 10-6 m/mK versus concrete → 7 ÷ 13 x 10-6 m/mK [Neville 1996: Properties of concrete, 4th edition]. CTE of carbon fibers (16% of the total bar volume), to some extent, affects the residual stresses inside the composite bar. Nevertheless, tests in fire conditions of beams on a natural scale (over 3m in length) carried out by me, indicate the beneficial effect of carbon fibers (in the appropriate amount) in the composite bar on the reduction of deflection and longer exposure to fire (which will be the subject of a separate work).

Point 3: Eq 1. suggest the ROM for tensile strength. However, this prediction is very inaccurate method with low. There are sevearal modified methods (ex, modification of rule of mixtures for tensile strength extimation of circular GFRP rebars, polymers, 9(12): 682, 2017) Plz check it.

Response 3:  Equation 1 was used to optimize the contribution of carbon and basalt fibers in the HFRP bar based on ROM for tensile strength. This was a preliminary approximation that was verified based on FEM analysis. The optimization goal was to increase the stiffness of the HFRP bar. The bar stiffness is to a large extent an immanent feature of the anisotropic material. I have read the indicated interesting article, for which I thank you [Modification of rule of mixtures for tensile strength extimation of circular GFRP rebars, Polymers, 9 (12): 682, 2017]. The shear lag phenomenon occurs mainly in materials consisting of fibers due to uneven stress. This phenomenon does not occur in steel due to its isotropic nature. The shear lag phenomenon occurs especially in anisotropic materials. The reduction in tensile strength is greatly influenced by the increase in bar diameter due to the shear lag phenomenon (Fig. 11 of the cited article). This phenomenon has also been confirmed in my previous research. As for determining tensile strength, the new ROM method is undoubtedly useful. However, in the case of optimization of the modulus of elasticity, depending on the fiber composition, the differences in elastic modulus can be determined using the current ROM method.

Point 4: Strength and stiffness should be constant regardless of the length. As the author mentioned, the load measured by the compression test (data of buckling zone in Figure 12) is a buckling load, not compressive strength. And the buckling load following the equation P = π2*E*I/L2. Only a short specimen (data of the crushing zone) is considered to be the compressive strength. Plz check the standard method to measure the compressive strength (ex, astm d695-10, d6641 or d3410)

Response 4:  The bars were tested for compression due to the changing length of the anchorage. Thank you for a good point about buckling. Under ideal real conditions, the bars always have some manufacturing inaccuracies, while the load can work obliquely or load the bars eccentrically. Under ideal real conditions, buckling caused by compression exists practically from the very beginning of the deformation. However, its effect is small until the values of forces become critical. As the length of the free anchored bars increases, the loss of stability occurs in the elastic working range of the material. For bars with heterogeneous HFRP fibers (with a matched k = 0.67), the curve determined according to the Euler equation (hyperbola) incorrectly maps the resistance to buckling load. As the free anchorage length increased, the load resistance decreased. Since the bars eventually buckled, one should speak about the resistance to buckling load. The ASTM D695-15 standard, which I used in the initial tests, limits its use to samples with a modulus of elasticity to 41.37 GPa (6000 ksi). Initially, I performed compression tests according to section 6.3 of the standard. Compressive strength for samples that were two diameters long was distorted due to local crushing of the bar ends. The use of anchorages has eliminated the impact of this phenomenon on the results of the measurement of strength on compressive load. For the modulus of elasticity, the standard recommends samples with four diameters. However, as my previous research has shown, it is necessary to properly install a strain gauge, which with smaller bar diameters is a great difficulty and can significantly affect the measurement result.

Point 5: Does the cylindrical jig of Figure 7-(a), not interfere with the buckling of the specimens?

Response 5:  The cylindrical device has sufficient width. During the tests, the bar behavior was carefully monitored and no impact of the steel sleeve on the course of the test was found.

Point 6: The actual rebar will not have a free length surrounded by concrete, what does it mean to take free length as a main test variable?

Response 6:  Naturally, under real conditions the bar is surrounded by concrete and there is no free length of the bar as such. If the distance between the stirrups holding the bar is large, the compressed bar will be buckled. As a consequence, the concrete cover may split off locally. This phenomenon has been observed during testing of bending beams, which I have already mentioned. There was a local spalling of the concrete cover and visible buckling of the bar in the compression zone. After the load ceased, due to the elastic behavior, the bar returned to its original position. Therefore, determining the proper free test length will allow to determine the optimal spacing of stirrups due to the required resistance to buckling load.

Point 7: It is judged that the slope of the strain varies depending on the applied stress as a result of the material being damaged or deformed. Can it be represent that the stiffness has changed as described in Figure 11.?

Response 7:  In composite materials there is a change in the modulus of elasticity (not very high) as the stress increases. A deformation in which stress and strain are proportional is called elastic deformation. The stress (ordinate) graph as a function of strain (abscissa) gives a linear relationship. However, there are materials (e.g. gray cast iron, concrete and many composites) for which this flexible part of the stress-strain curve is not linear.For this non-linear behavior, the tangent or secant modulus of elasticity is usually used. The tangent modulus is taken as the slope of the stress-strain curve at a given stress level, while the secant modulus represents the slope of the secant drawn from the beginning to a certain point on the curve. This incisal modulus assumption was adopted in composites standards (e.g. ACI 440 and CSA S806 standards). It was assumed that the modulus of elasticity should be determined as the ratio of stress difference to deformation difference for the same range of loads (0.2 ÷ 0.5Fu).

Figure 11 shows the relationship of the tangential modulus of elasticity on stress. According to ACI440.3R-15, the modulus should be determined as the ratio of stress difference to deformation difference for the same load range (0.2 ÷ 0.5Fu). Thus, it is the angle of inclination (or more precisely the tangent of the angle) of the incisal stress-strain curve. In contrast, the "local" variability of the modulus of elasticity (understood as the ratio of successive stresses to subsequent deformations) slightly changes with increasing load on the bar. The variability of the tangent modulus of elasticity in the above range for tension bars is about 2.8%. However, for compressed bars the variability is much smaller and amounts to about 0.5%.

All subsequent changes are detailed below in the cover letter:

  1. The chapter: “Characteristics of basalt and carbon fibers. has been moved: 72 line.
  2. Sentence added: “Bar Hybridization” : 93 line.
  3. Sentence added: “The purpose of the new approach to assessing the compressive strength of composite bars is to use them as reinforcement in compressed RC elements (columns) as well as in the compressed zone of bent elements (e.g. beams and slabs).” : 52 line.
  4. The sentence has been moved: “The properties of composite materials, including elasticity modules, were calculated on the basis of the mixture rule (ROM) [19-20].” : 103 line.

    5. Sentence added: “Failure Mechanism”  235 line.

  1. Sentences added : “Microscopic Analysis. Before starting the compression tests, it was decided to conduct a microscopic analysis to determine the effect of chemical composition on the properties of HFRP bars.” : 253 line.
  2. Sentences added: “In composite materials, the modulus of elasticity (not very large) changes as the stress increases. For this non-linear behavior, the tangent or secant modulus of elasticity is usually used.” : 402 line.
  3. Sentences added: “Under ideal real conditions, the bars always have some manufacturing inaccuracies, while the load can work obliquely or load the bars eccentrically. Under this conditions, buckling caused by compression exists practically from the very beginning of the deformation. However, its effect is small until the values of forces become critical. As the length of the free anchored bars increases, the loss of stability occurs in the elastic working range of the material.” 450 line.
  4. Sentence added: “The stress - strain relationship, shown in Figure 12, is linear in all three of these zones and is well corresponded to the test results.” : 471 line.
  5. The word “compression” was changed to “buckling load” at line: 21, 281, 284, 287, 296, 336, 338, 368, 373, 378, 379, 380, 382, 447, 448, 450, 455, 457, 462, 464, 466, 473, 477, 484, 494, 500.
  6. The word “tangent” has been added at line: 421, 427, 430, 431, 433, 435, 436, 440, 442, 443, 488, 489, 492, 493, 496.

Reviewer 2 Report

This article deals with the development of new experimental device to evaluate the compressive properties of bar made with composite materials. The author has performed tests on Hybrid Fiber Reinforced Polymer Hydrib bars. The term hybrid means that several types of fibers are used. Carbon and basalt fibers are incorporated in epoxy matrix with the following respective volume fractions (16%/64%20%). The main objective of the article seems to be to propose a new approach to evaluate the compressive strength of composite bars in the final goal to use them as reinforcement in concrete columns. For me, the novelty and the main objective of the paper have to be clearly defined.

The paper is not well written.

  • Some parts of the paper seem to be glued in the document without link with the previous part (example : page 7 paragraph line 252 to 259)
  • The order of paragraphs is not logical (Characteristics of basalt and carbon fibers before the Novelty and Purpose of the Work)
  • The same think is observed inside the paragraph : Equations of the mixture rule are written line 100 and line 111 the author writes “The properties of composite materials, including elasticity modules, were 111 calculated on the basis of the mixture rule (ROM)”

The major problem concerns the study itself. How can we define a compressive strength (which is an intrinsic property) depending of the length of the specimen? All the standards impose a maximum length (more precisely a ratio diameter on length) to avoid buckling. If the buckling occurs, the test is not valid and no results have to be post-treated. Concerning the calculation of the elastic modulus, the most important thing is to check that it is made in the linear part of the strain-stress curve. Figure 11 shows the evolution of the elastic modulus with the stress. If it is not constant this means that the strain-stress curve is not linear and so the graph shows the evolution of the tangent modulus.

I recommend revision in depth of the manuscript, addressing the comments, to make this a paper suitable for publication in Materials.

Author Response

At the beginning I would like to thank you for the insightful and accurate comments contained in the commentary to the article. Below I will try to answer the issues raised and also adapt to the comments in the article.

Point 1: The main objective of the article seems to be to propose a new approach to evaluate the compressive strength of composite bars in the final goal to use them as reinforcement in concrete columns. For me, the novelty and the main objective of the paper have to be clearly defined.

Response 1: According to the suggestions, I placed the chapter on novelty and purpose of the work before (characteristics of basalt and carbon fibers. I put the following sentence specifying the main purpose of the work: "The purpose of the new approach to assessing the compressive strength of composite bars is to use them as reinforcement in compressed RC elements (columns) as well as in the compressed zone of bent elements (e.g. beams and slabs)."

Point 2: Some parts of the paper seem to be glued in the document without link with the previous part (example : page 7 paragraph line 252 to 259)

Response 2: I decided to extract a fragment of the text on page 7 by the subtitle "Microscopic analysis". Thank you for indicating no link to the previous part of the article. I decided to introduce the following explanation sentence at the beginning of this subsection: "Before proceeding to compression tests, it was decided to carry out microscopic analysis to determine the effect of chemical composition on the properties of HFRP bars."

Point 3: The order of paragraphs is not logical (Characteristics of basalt and carbon fibers before the Novelty and Purpose of the Work)

Response 3: According to the suggestions, I placed the chapter on novelty and purpose of the work before (characteristics of basalt and carbon fibers.

Point 4: The same think is observed inside the paragraph : Equations of the mixture rule are written line 100 and line 111 the author writes “The properties of composite materials, including elasticity modules, were 111 calculated on the basis of the mixture rule (ROM)”

Response 4:  In Chapter 3, according to the remark, I moved the following sentence before Equation 1: “The properties of composite materials, including elasticity modules, were calculated on the basis of the mixture rule (ROM) [19-20]” .

Point 5: The major problem concerns the study itself. How can we define a compressive strength (which is an intrinsic property) depending of the length of the specimen? All the standards impose a maximum length (more precisely a ratio diameter on length) to avoid buckling. If the buckling occurs, the test is not valid and no results have to be post-treated.

Response 5:  Compressive strength problem: Indeed, a distinction must be made between compressive strength and resistance (resistance) to buckling load on bars. The bars were tested for buckling load due to the changing anchor length. As the free anchoring length increases, the load resistance decreases due to the increasing buckling effect. Since the bars eventually buckled, one should speak about the resistance to buckling load.

Point 6: Concerning the calculation of the elastic modulus, the most important thing is to check that it is made in the linear part of the strain-stress curve. Figure 11 shows the evolution of the elastic modulus with the stress. If it is not constant this means that the strain-stress curve is not linear and so the graph shows the evolution of the tangent modulus.

Response 6:  Modulus of elasticity problem: In composite materials, the modulus of elasticity (not very large) changes as the stress increases. I agree that the graph shows the evolution of the tangential modulus of elasticity. A deformation in which stress and strain are proportional is called elastic deformation. The stress (ordinate) graph as a function of strain (abscissa) gives a linear relationship. However, there are materials (e.g. gray cast iron, concrete and many composites) for which this flexible part of the stress-strain curve is not linear.

For this non-linear behavior, the tangent or secant modulus is usually used. The tangent modulus is taken as the slope of the stress-strain curve at a given stress level, while the secant modulus represents the slope of the secant drawn from the beginning to a certain point on the curve. This incisal modulus assumption was adopted in composites standards (e.g. ACI 440 and CSA S806 standards). It was assumed that the modulus of elasticity should be determined as the ratio of stress difference to deformation difference for the same range of loads (0.2 ÷ 0.5Fu).It should be noted that at the atomic scale, macroscopic elastic deformation manifests itself as small changes in interatomic spacing and stretching of interatomic bonds. Consequently, the size of the modulus of elasticity is a measure of the resistance to separation of adjacent atoms, i.e. the forces of interatomic bond. The modulus of elasticity for composite materials is smaller compared to metals. These differences are a direct consequence of different types of atomic bonds in two types of materials. Thank you for pointing out that Figure 11 shows the evolution of the tangent modulus of elasticity, which was reviewed in the article.

All subsequent changes are detailed below in the cover letter:

  1. The chapter: “Characteristics of basalt and carbon fibers. has been moved: 72 line.
  2. Sentence added: “Bar Hybridization” : 93 line.
  3. Sentence added: “The purpose of the new approach to assessing the compressive strength of composite bars is to use them as reinforcement in compressed RC elements (columns) as well as in the compressed zone of bent elements (e.g. beams and slabs).” : 52 line.
  4. The sentence has been moved: “The properties of composite materials, including elasticity modules, were calculated on the basis of the mixture rule (ROM) [19-20].” : 103 line.

    5. Sentence added: “Failure Mechanism”  235 line.

  1. Sentences added : “Microscopic Analysis. Before starting the compression tests, it was decided to conduct a microscopic analysis to determine the effect of chemical composition on the properties of HFRP bars.” : 253 line.
  2. Sentences added: “In composite materials, the modulus of elasticity (not very large) changes as the stress increases. For this non-linear behavior, the tangent or secant modulus of elasticity is usually used.” : 402 line.
  3. Sentences added: “Under ideal real conditions, the bars always have some manufacturing inaccuracies, while the load can work obliquely or load the bars eccentrically. Under this conditions, buckling caused by compression exists practically from the very beginning of the deformation. However, its effect is small until the values of forces become critical. As the length of the free anchored bars increases, the loss of stability occurs in the elastic working range of the material.” 450 line.
  4. Sentence added: “The stress - strain relationship, shown in Figure 12, is linear in all three of these zones and is well corresponded to the test results.” : 471 line.
  5. The word “compression” was changed to “buckling load” at line: 21, 281, 284, 287, 296, 336, 338, 368, 373, 378, 379, 380, 382, 447, 448, 450, 455, 457, 462, 464, 466, 473, 477, 484, 494, 500.
  6. The word “tangent” has been added at line: 421, 427, 430, 431, 433, 435, 436, 440, 442, 443, 488, 489, 492, 493, 496.

Round 2

Reviewer 1 Report

As for determining tensile strength, the new ROM method is undoubtedly useful. However, in the case of optimization of the modulus of elasticity, depending on the fiber composition, the differences in elastic modulus can be determined using the current ROM method.

-> That was why recommoned deleting the Eq (1) ROM for strength, which is inaccurate and unnecessary for this paper.

Figure 11 shows the relationship of the tangential modulus of elasticity on stress. According to ACI440.3R-15, the modulus should be determined as the ratio of stress difference to deformation difference for the same load range (0.2 ÷ 0.5Fu). Thus, it is the angle of inclination (or more precisely the tangent of the angle) of the incisal stress-strain curve. In contrast, the "local" variability of the modulus of elasticity (understood as the ratio of successive stresses to subsequent deformations) slightly changes with increasing load on the bar. The variability of the tangent modulus of elasticity in the above range for tension bars is about 2.8%. However, for compressed bars the variability is much smaller and amounts to about 0.5%.

-> Following this express, the slope of tensile SS curve increased. Is is possible to increase the slope (stiffness) even considering work hardning (strain hardning)? Please present the SS curve of tensile test to make clear the data.

Author Response

Point 1: As for determining tensile strength, the new ROM method is undoubtedly useful. However, in the case of optimization of the modulus of elasticity, depending on the fiber composition, the differences in elastic modulus can be determined using the current ROM method.

-> That was why recommoned deleting the Eq (1) ROM for strength, which is inaccurate and unnecessary for this paper.

Response 1: Thank you for paying attention to the oversight. The equation due to ROM regarding composite bar strength has been removed (after 146 line).

Point 2: Figure 11 shows the relationship of the tangential modulus of elasticity on stress. According to ACI440.3R-15, the modulus should be determined as the ratio of stress difference to deformation difference for the same load range (0.2 ÷ 0.5Fu). Thus, it is the angle of inclination (or more precisely the tangent of the angle) of the incisal stress-strain curve. In contrast, the "local" variability of the modulus of elasticity (understood as the ratio of successive stresses to subsequent deformations) slightly changes with increasing load on the bar. The variability of the tangent modulus of elasticity in the above range for tension bars is about 2.8%. However, for compressed bars the variability is much smaller and amounts to about 0.5%.

-> Following this express, the slope of tensile SS curve increased. Is is possible to increase the slope (stiffness) even considering work hardning (strain hardning)? Please present the SS curve of tensile test to make clear the data.

Response 2: The purpose of Figure 11 was to show whether and how the tangential modulus of elasticity changes depending on the stress caused by the load. All stress - strain curves in HFRP bar tensile tests are linear. During the tensile tests, I conducted 105 tests only for the HFRP bars themselves (for different bar diameters). Changing the value of tangent tensile modulus of elasticity in the load range (0.2 ÷ 0.5 Fu) of only 2.8% value, there will be noticeable on the commonly used stress – strain relationship.

The curves in Fig. 11 are not stress - strain relationship commonly used to determine the modulus of elasticity. However, they describe the tangent modulus of elasticity - stress (load). The falling curves (red line) relate to the tangent compression modulus of elasticity - compressive stress relationship. In contrast, rising curves (blue line) relate to the tangent tensile modulus of  elasticity - tensile stress relationship. Figure 11 gives the absolute stress values (without taking into account the sign, i.e. whether they are tensile or compressive). In this way, the change in value (relatively small - as I have already written) of the tangent modulus of elasticity in compression and tension is compared. In order to clearly define the type of modulus of elasticity in the sentence: "In contrast, the" local "variability of the modulus of elasticity (understood as the ratio of successive stresses to subsequent deformations) slightly changes with increasing load on the bar." before "modulus of elasticity" should be "tangent".

Reviewer 2 Report

The author has taken into account the comments and the remarks of the reviewer. The quality of the manuscript has been improved.

I have still 2 remarks :

  • How does the author measure the section area of the bar? In figure 8(b), we can see that the section is not constant along the bar.
  • I am not agree with the sentence "In composite materials, the modulus of elasticity (not very large) changes as the stress increases.". In the case of continuous fiber reinforced composite materials, the elastic modulus can be large and the linear part (where the elastic modulus is constant) of the stress-strain curve is not negligible.

Author Response

Point 1: How does the author measure the section area of the bar? In figure 8(b), we can see that the section is not constant along the bar.

Response 1: In order to determine the reliable cross-section of the bar during the compression test, the findings included in standard D695-15 were used. Compressive (nominal) stress is defined as the compressive load per unit area with a minimum original cross-section within the measuring instrument, transferred by the sample under test at a given moment. The measurement of the minimum diameter of the "naked" bar (without taking into account the braid) was made in at least three places between the anchorages. Measurement was carried out three times in each place, each time around a longitudinal axis every 120 degrees, with an accuracy of  0.1 mm.

However, during the tensile test, due to the varied form of the bars, e.g. ribbing or braid of composite fibers - it was necessary to determine the equivalent values: diameter, cross-sectional area and circumference. The above values were determined by measuring the volume of water displaced by FRP bar samples in a graduated cylinder of appropriate volume and length. The measurement of the length of each bar was made three times, each time around the longitudinal axis every 120 degrees, with an accuracy of 0.1 mm. The cross-sectional area of the bars is determined with an accuracy of 1 mm2 by dividing the volume by the length of the bar. Tensile stress is the tensile load per unit area with the above described cross section (which is in accordance with the ACI 440.3R procedure, CSA S806, ASTM D7205).

Point 2:    I am not agree with the sentence "In composite materials, the modulus of elasticity (not very large) changes as the stress increases.". In the case of continuous fiber reinforced composite materials, the elastic modulus can be large and the linear part (where the elastic modulus is constant) of the stress-strain curve is not negligible.

Response 2: Thank you for pointing out the wrong wording. The modulus of elasticity under compression should be defined as the ratio of stress (nominal) to the corresponding deformation below the material proportionality limit. I accepted the sentence: “„In the case of continuous fiber reinforced composite materials, the elastic modulus can be large and the linear part (where the elastic modulus is constant) of the stress-strain curve is not negligible.”